

# Fake news detection: state-of-the-art review and advances with attention to Arabic language aspects

Eman Salamah Albtoush[1], Keng Hoon Gan[1] and
Saif A. Ahmad Alrababa[2]

[1] School of Computer Sciences, Universiti Sains Malaysia, Gelugor, Malaysia
[2] Faculty of Information Technology, Al al-Bayt University, Mafraq, Jordan

## ABSTRACT

The proliferation of fake news has become a significant threat, influencing individuals, institutions, and societies at large. This issue has been exacerbated by the pervasive integration of social media into daily life, directly shaping opinions, trends, and even the economies of nations. Social media platforms have struggled to mitigate the effects of fake news, relying primarily on traditional methods based on human expertise and knowledge. Consequently, machine learning (ML) and deep learning (DL) techniques now play a critical role in distinguishing fake news, necessitating their extensive deployment to counter the rapid spread of misinformation across all languages, particularly Arabic. Detecting fake news in Arabic presents unique challenges, including complex grammar, diverse dialects, and the scarcity of annotated datasets, along with a lack of research in the field of fake news detection compared to English. This study provides a comprehensive review of fake news, examining its types, domains, characteristics, life cycle, and detection approaches. It further explores recent advancements in research leveraging ML, DL, and transformer-based techniques for fake news detection, with a special attention to Arabic. The research delves into Arabic-specific pre-processing techniques, methodologies tailored for fake news detection in the language, and the datasets employed in these studies. Additionally, it outlines future research directions aimed at developing more effective and robust strategies to address the challenge of fake news detection in Arabic content.

# INTRODUCTION

Fake news refers to intentionally created media content that uses manipulative techniques to imitate authentic and trustworthy news sources with the purpose of misleading the reader (*Tandoc, Lim & Ling, 2018*). Recently social media platforms have been considered the sources for fake news dissemination (*Collins et al., 2021*). As of early April 2024, there are 5.07 billion social media users worldwide (https://datareportal.com/social-media-users). Statistic indicates that 86% of global citizens initially believe news published through social media without verifying its truthiness. Additionally, 67% of individuals report encountering fake news on Facebook, 60% on websites, 56% on YouTube, and 51% on television (*Simpson, 2019*). The ease of publishing, sharing, and commenting or

Corresponding author
Eman Salamah Albtoush,
emanbtoush@student.usm.my

tweeting without prior verification of authenticity has flooded the digital world with a massive amount of unreliable data, facilitated by its unrestricted and cost-free accessibility. Recently, many examples of fake content about the COVID-19 pandemic (*Akhter et al., 2024*), political agendas like elections (*Pennycook & Rand, 2021a*), and economic threats (*Olan et al., 2024*), demonstrate how fake news spreads with the help of social media. However, beyond mimicking news media, there is often an underlying intent driving and motivate the propagation of fake news. Fake news aims to mislead and deceive readers for political or financial gain. Figure 1 illustrates the motives that drive the dissemination of fake news across digital social platforms. Fake news can be classified into several types, including: satirical or parodic content intended to entertain the audience with a humor rather than deceive them, such as Satirewire (https://www.satirewire.com) and The Onion (https://www.theonion.com/) (*Shu et al., 2017*). Other examples of ways to spread fake news include: clickbait, which relies on completely fake information to grab the attention of readers and generate profits through deceptive advertisements, reviews, and fake headlines, such as "yellow journalism" (*Shu et al., 2017*); propaganda, which manipulates people's beliefs, emotions, and opinions for political or ideological reasons, often through selective news framing (*De Vreese, 2005*) and conspiracy theories (*Tandoc, Lim & Ling, 2018*); hoaxes involve fabricating and purposefully spreading untruthful content for entertainment, which spreads quickly to create significant impacts, such as HoaxSlayer (https://www.hoax-slayer.com/); and rumors, which are unverified and exaggerated claims. Figure 2 illustrates various fake news categories according to content and intent.

There are various ways to mitigate the effects of widespread fake news, including the efforts of traditional fact-checking organizations and social media platforms. These organizations play a crucial role in identifying fake content through extensive efforts, whether *via* manual processes or algorithmic approaches. For example, platforms like Twitter Crawler" (https://x.com/crawlershq) and "Streaming API" (*De Beer & Matthee, 2021*) are used to collect tweets in a database, which are then checked by users to ensure their accuracy (*Atodiresei, Tănăselea & Iftene, 2018*). Additionally, X, formerly known as Twitter, has added an "ASK EXPERTS" link to connect users with credible sources to verify their news. WhatsApp, for instance, considered a main source of spreading rumors related to COVID-19, recently introduced the phrase "Forwarded many times" on messages that have been frequently forwarded to alert users that the content may not be accurate. Facebook reduces the visibility of potential fake news articles and provides users with instructions to overcome misinformation (*Sparks & Frishberg, 2020*). Instagram similarly directs users searching for credible information (*Marr, 2020*). Online fact-checking tools such as Fullfact (https://fullfact.org/), Logically (https://www.logically.ai/), TruthOrFiction (https://www.truthorfiction.com/), and Reporters Lab (https://reporterslab.org/) are widely used to detect and verify fake content globally. Fatabyyano (https://rb.gy/fdf98v) and Misbar (https://misbar.com/), both tailored for Arabic, focus specifically on detecting and verifying fake content in the Arabic-speaking world. However, traditional methods like fact-checking, which rely on knowledge and expertise, are time-consuming and tedious. They also suffer from limited scalability, reliance on human resources, and delays in addressing rapidly spreading fake news. These limitations

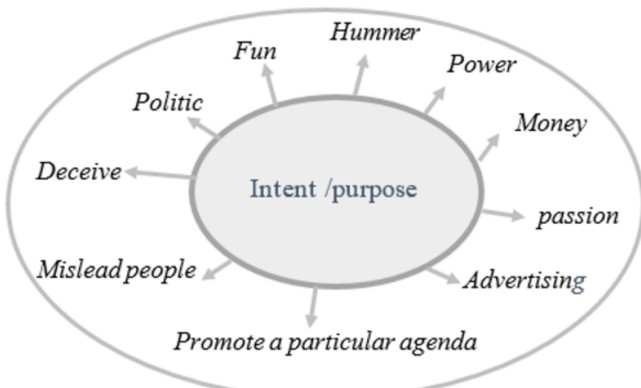

**Figure 1 Motivations behind fake news and the diverse types of information on online social networks.**

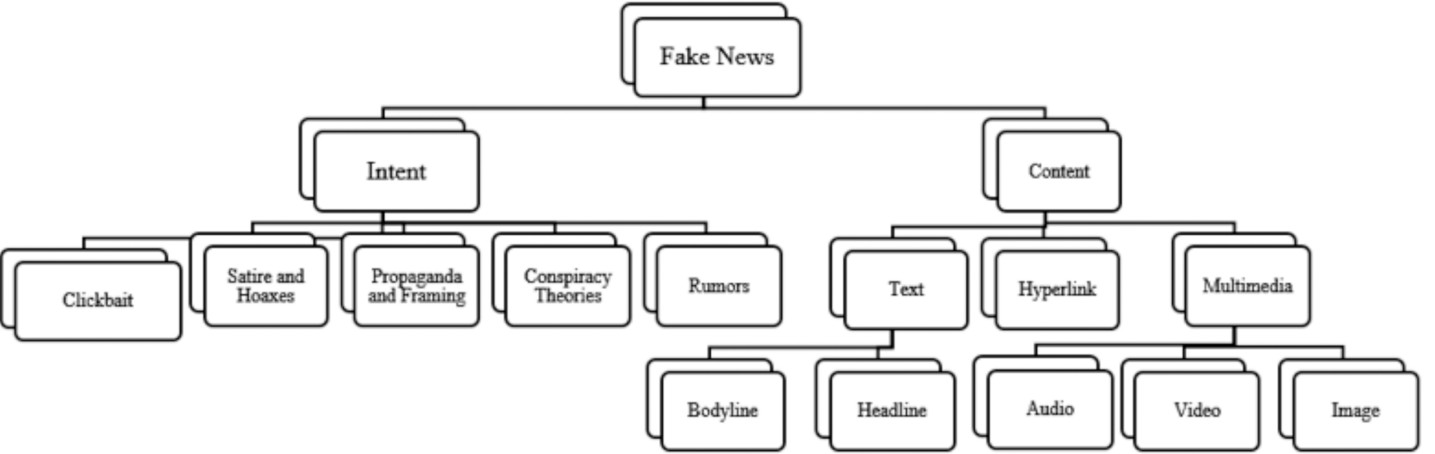

**Figure 2 Fake news types categorized based on intent and content type, the focus of this review.**

underscore the importance of developing automated and efficient systems to counter fake news effectively. Assisting and supporting individuals in identifying fake news is fundamental to the success of any fake news detection system. Machine learning offers a promising approach for spotting fake news, as it is cost-effective and requires fewer human resources. This study reviews various fake news detection methods, including machine learning (ML), deep learning (DL), and Transformer-based approaches, with a particular focus on their application to the Arabic language. Interest in processing Arabic has been growing recently, given that it is spoken by over 400 million people across 22 countries worldwide (*Boudad et al., 2018*). People in various Arab regions report significant exposure to fake news. In Lebanon, 88.1% of people encounter fake news, while in Saudi Arabia, the figure is 85%. In Qatar, 74.3% of the population reports encountering fake news, followed by 68.7% in the United Arab Emirates and 66.6% in Tunisia (*Martin & Hassan, 2020*). Fake news about the COVID-19 pandemic and Arab Spring (*Alsafadı, 2023*) are examples of fake information that significantly affect the Arab region. The challenges in combating

fake news are further compounded by the linguistic and semantic complexities of the Arabic language, as well as the scarcity of reliable sources and comprehensive datasets that address multi-domain aspects, which are often reiterated throughout the narrative (*Shaalan et al., 2019*). This study targets a broad audience, including researchers, data scientists, and professionals focused on fake news detection, ML, and social media analysis. The motivation behind this work is to build a high-quality information base to aid researchers in advancing Arabic fake news detection. It identifies state-of-the-art ML techniques and offers practical insights into addressing gaps, such as dataset scarcity. The article also presents specialized solutions and recommendations, covering dataset challenges and strategies for improvement in Arabic content analysis. The review includes the following sections: "Overview of Fake News" offers an overview of fake news, "Fake News Detection Approaches" delves into various detection approaches, "Machine Learning Techniques to Detect Fake News" explores techniques for detecting fake news, "Fake News Detection Challenges" addresses the challenges encountered in detecting fake news, "Future Research Direct" outlines future research, and "Conclusion" presents the review conclusion.

## Search and survey methodology summary

The following search phrases were used to find literature from 2020 to 2024: "Fake news", "Fake news approaches", "Fake news life cycle", "Fake news detection", "Arabic fake news detection", "Fake news Datasets", "Optimal fake news dataset" and "Machine learning methods for Fake news detection". The search was conducted on six scientific databases: IEEE, Springer Link, Taylor & Francis Online, ACM digital library, MDPI, and Elsevier. These databases were chosen considering their extensive coverage and relevance in the domain of computer science. Google Scholar was also searched to increase publication coverage. Search results were filtered based on the following criteria: articles had to be peer-reviewed, published from 2020 onward, and relevant to fake news detection. Relevancy was assessed by reviewing the abstracts, methodologies, and findings to ensure the inclusion of studies introducing novel datasets, advanced detection techniques, or addressing issues specific to Arabic content. Duplicate results were removed, and articles with limited applicability to fake news detection were excluded. In order to maintain focus on the most recent advancements, peer-reviewed publications published in 2020 and later were prioritized over earlier or irrelevant research.

## Research questions

- **RQ1**: What domains and traits define fake news, and how do they influence its spread on social media?
- **RQ2**: How can understanding the fake news life cycle enhance detection methods?
- **RQ3**: How do linguistic, topic-agnostic, visual, and social approaches compare in detecting fake news, and what are their strengths and limitations?

**Table 1 Sample of fake news and its domain.**

| Domain | Event | Source |
|---|---|---|
| Natural disaster | - A nuclear explosion and the Chilean government were blamed for the earthquake in Chile 2010. | *Mendoza, Poblete & Castillo (2010)* |
| | - Turkey–Syria earthquake misinformation. | *Méndez-Muros, Alonso-González & Pérez-Curiel (2024)* |
| Health | - COVID-19 misinformation (https://rb.gy/j96q4c). | *Nie (2020)* |
| | - Misinformation about the power of salt related to the nuclear crisis in Japan in 2011. | *Guo (2020)* |
| Economic | - False reports tanked United Airlines stock. | *Chen et al. (2019)* |
| | - Brexit misleading news. | *Greene, Nash & Murphy (2021)* |
| Politics | - Misleading content about the Brazilian and Indian elections. | *Reis et al. (2020)* |
| | - Egyptian-Mexican relations in 2015. | *Saadany, Mohamed & Orasan (2020)* |

- **RQ4**: How do social context and user behavior impact Arabic fake news detection?
- **RQ5**: Which machine learning models and datasets are most effective for detecting fake news in English and Arabic?
- **RQ6**: Do pre-processing techniques improve Arabic fake news detection accuracy?
- **RQ7**: What challenges exist in building reliable datasets for Arabic fake news detection?
- **RQ8**: How can early fake news prediction be improved, and which features are most influential?
- **RQ9**: What unique challenges does Arabic pose for fake news detection, and how can they be overcome?

## OVERVIEW OF FAKE NEWS

### Domains of fake news

The propagation of misleading content spans various domains, including politics, health, art, economy, and entertainment. Fake news may be more prominent in certain fields, such as politics, but its impact remains significant across all domains. Globally, events like COVID-19 and the 2016 U.S. presidential election have heightened the focus on detecting fake news. For instance, misleading content about global conflicts, such as the wars between Russia and Ukraine (*Shin et al., 2023*), the recent Hamas-Israel conflict (*Shahi, Jaiswal & Mandl, 2024*), the Facebook revolution (https://rb.gy/307rns) in the Arab region following the Arab Spring, and various other global events, underscores the widespread impact of misleading information. Table 1 shows examples of fake news and their respective domains. One recent example, illustrated in Fig. 3, involves fake news shared on a Facebook profile (http://tiny.cc/4toa001) about a genuine earthquake in the Arab region in 2023. The post falsely claimed that the "British Meteorological Center" predicted a 6.2-magnitude earthquake in the Arabian Gulf region and Yemen in the coming hours. Fatabyyano (https://rb.gy/fdf98v), a popular Arabic fact-checking organization, debunked this fabricated post.

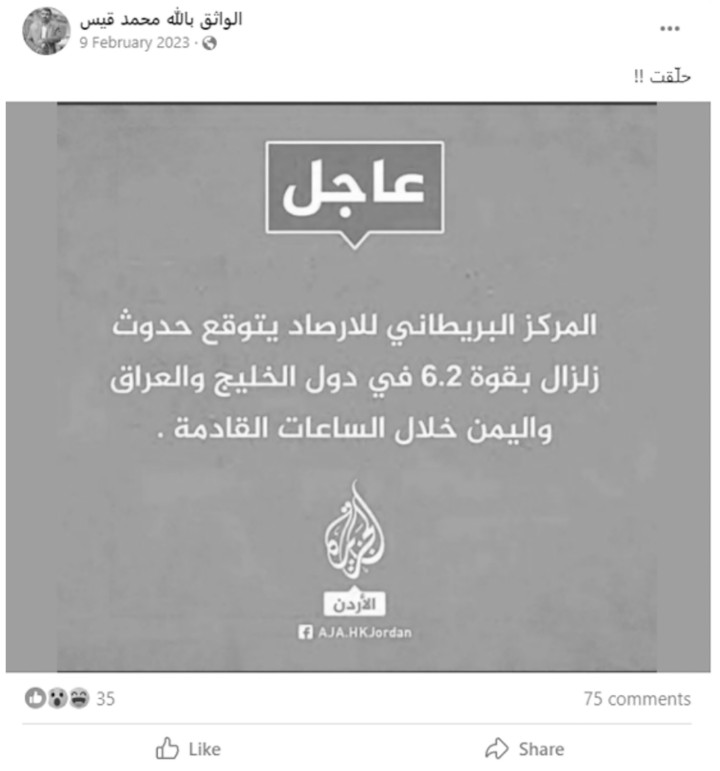

**Figure 3 Example of Arabic fake news: Facebook post fabricating an earthquake in the Arabian region.**

## Characteristics of fake news

Understanding the characteristics of misleading content propagation is crucial for individuals and governments to control, target, and reduce its impact. Especially with the widespread and intense use of social media, key characteristics of untruth news include: intention to deceive (*Thota et al., 2018*), sensationalist headlines with exaggerated or unbelievable claims (*Wei & Wan, 2017*), echo chambers reinforcing existing beliefs, lack of credible sources (*Shao et al., 2018*), rapid spread (*Vosoughi, Roy & Aral, 2018*), manipulated content, poor writing quality, biased writing supporting a particular point of view or agenda (*Pennycook & Rand, 2021b*), and distinct linguistic features, such as the use of more proper nouns, comparatives, conjunctions, and adverbs. For instance, word pairs, rather than single words, can better indicate fake news due to the significant role of semantic properties (*Shrestha & Spezzano, 2021*).

## Fake news life cycle

It is important first to understand the fake news life cycle: creation, dissemination, early detection, and propagation. The way users interact, respond, share, support, or ignore news tainted with falsehoods is crucial. Figure 4 illustrates the fake news life cycle, spanning four stages: *Creation* involves fabricated, inaccurate, or misleading content to manipulate or entertain people through sensationalist or exaggerated headlines. *Dissemination* refers to the spread of fake content through various traditional and, more

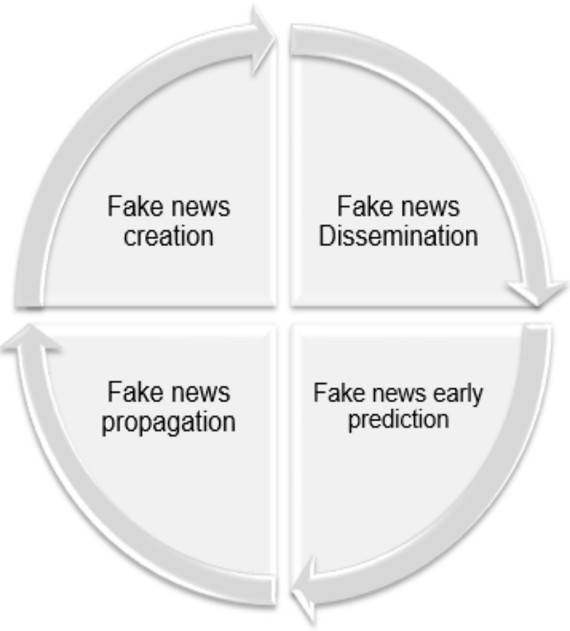

**Figure 4  Life cycle of fake news.**

significantly, social media platforms, which facilitate the sharing of misleading content. *Early Detection* aims to prevent widespread *propogation* and minimize its impact based on real-time analysis (*Zubiaga et al., 2016*). A multimodal approach for early fake news detection was proposed, addressing the limitations of previous methods that relied on specific models. This approach focused on the characteristics of fake news content and propagation data, utilizing graph neural networks (GNNs) and BERT (Bidirectional Encoder Representations from Transformers). By extracting propagation patterns and analyzing news content for social media posts, the method achieved impressive results on public datasets (*Sormeily et al., 2024*). In contrast, research in the field of early fake news detection in Arabic is considered rare, with only a few projects, such as those recently presented by Qatar University, supporting this area (https://n9.cl/6oifi).

## FAKE NEWS DETECTION APPROACHES

### Knowledge-based approach

The knowledge-based approach is widely used for verifying the truthfulness of news by consulting experts, professionals, crowdsourced (https://www.fiskkit.com/), and fact-checking websites or organizations. Many fact-checking sites employ manual detection, such as FactCheck (https://www.factcheck.org/) and Politifact (https://www.politifact.com/). In the Arab world, for instance, several organizations have been founded for fact-checking, particularly in the fields of politics and health, such as Fatabyyano. Other organizations, like DaBegad (https://dabegad.com/), Matsda2sh (https://matsda2sh.com/), and Misbar, have also found developing automated techniques for fake news detection necessary, although these sites still operate primarily based on manual detection (*Alkhair et al., 2019*). Table 2 provides examples of popular online fact-checking

**Table 2 Fact-checker organizations and websites, their languages, and detecting techniques.**

| Name | Website | Language | Detecting technique |
|---|---|---|---|
| Factcheck | https://www.factcheck.org/ | English | Manual |
| Originiality.ai | https://originality.ai/automated-fact-checker | English | Automated |
| Kashif | https://kashif.ps/ | Arabic | Manual |
| Arabfcn | https://arabfcn.net/ | Arabic | Manual |
| Norumors | http://norumors.net/ | Arabic | Manual |
| Verify-sy | https://verify-sy.com/ | Arabic | Manual |
| Tanbih | https://tanbih.qcri.org/ | Arabic | Manual |
| Snopes | https://www.snopes.com/ | English | Automated |
| Factcheck | https://factcheck.afp.com/ | Multi-language | Manual |
| Factmata | https://factmata.com/ | English | Automated |
| Maharat-news | https://maharat-news.com/fact-o-meter | Arabic | Manual |

organizations in English and Arabic. However, manual fact-checking is characterized by several key aspects. It is time-consuming, requires daily updates on ongoing news, and is subject to bias and subjectivity, such as in crowdsourced fact-checking, which is less reliable.

## Linguistic approach

This approach focuses on grammar, syntax, and semantics using various computational techniques to analyze and understand human language. It analyzes linguistic patterns such as specific writing patterns in news content, sentiment patterns and word frequencies, stylometric features, lexicon textual characteristics (*Zhang & Ghorbani, 2020*), syntax, semantics, discourse levels and grammatical structure to detect anomalies in the data. Features that can be extracted based on the language approach include lexical features such as character and word counts, large and unique words, sentence features, phrases, punctuation, and parts of speech (POS) tagging. For example, extracting lexical features from Arabic textual content has yielded promising results, with 78% accuracy in identifying fake news in a study by *Himdi et al. (2022)*. However, the linguistic approach has limitations in Arabic due to its rich morphology, diverse dialects, and lack of standardized spelling, particularly in dialectal texts. These challenges hinder feature extraction, complicate generalization, and impede the detection of nuanced semantics or authentic mimicry, reducing its scalability and robustness for fake news detection (*Shaalan et al., 2019*; *Saadany, Mohamed & Orasan, 2020*).

## Topic-agnostic approach

In the topic-agnostic approach, the focus is more on identifying and flagging fake news regardless of the specific topic or content, such as trustworthiness of the news source, the writing style and tone, and, moreover, the presence of certain patterns or elements meant to elicit emotional responses (*Horner et al., 2021*). Psychological features, such as those

related to social interactions or family and friends, and biological process features, such as health or sexual references, can be analyzed to address fake news. Presence of an author's name, long words, and URLs are also considered as core elements for assessing source credibility, writing style which involve techniques used to create highly biased and one side news include: emotional language, selective facts, exaggeration, attacks against opposing people views (*Hoy & Koulouri, 2021*). However, when applied to the Arabic language, this approach faces challenges due to dialectal variations and ambiguous writing styles. The meaning is often concealed behind words, with shifts in tone and lack of clarity. The use of emotional cues and diverse writing styles across dialects further complicates the identification of patterns indicative of fake news (*Shaalan et al., 2019*).

## Visual-based approach

Visual material can significantly enhance the credibility of a news article; textual content is not sufficient when used alone. Visual statistical modeling techniques and statistical features effectively assess news credibility. An overview of fake image detection is discussed, along with benchmarking tools by *Sharma et al. (2023)*. Deep fake images are one of the key challenges of this approach, as they require specialized algorithms to analyze metadata, content, lighting, shadows, and facial expressions (*Yu et al., 2021*).

## Social context approach

The social context approach relies on the social context of misleading content along with network-based features. Profile characteristics such as verified status, number of followers, account age, user behavior (including frequency of posts, retweet patterns, user interactions), temporal patterns (such as time of posting, temporal spread patterns), and integration types (including comments, likes, and shares) are considered. Additionally, it explores the propagation path of misleading content, such as content and contextual information, as investigated by *Passaro et al. (2022)*. Various social media elements were introduced and investigated by *Nielsen & McConville (2022)*. However, changes in user behavior, the nature of noisy data, account characteristics and their credibility, the propagation mechanisms of fake news, its ambiguity, the volume of real-time news, and its multilingual characteristics are all challenges faced by this approach (*Shu, Wang & Liu, 2019*; *Shu et al., 2020a*).

For Arabic fake news detection, a multi-approach strategy can be used. Knowledge-based systems could integrate Arabic fact-checking organizations for quicker verification (*Murayama, 2021*), while linguistic models analyze unique Arabic linguistic markers, such as emotional tone (*Hamed, Ab Aziz & Yaakub, 2023*). Topic-agnostic methods assess credibility across topics by tracking authorship and style patterns (*Liu et al., 2021*), visual-based tools can verify Arabic visuals and detect manipulated images (*Giachanou, Zhang & Rosso, 2020*), and social context analysis would examines Arabic social media behaviors, focusing on user profiles and interaction patterns to identify and track fake news spread (*Shu, Wang & Liu, 2019*), Fig. 5 illustrates these approaches and their primary applications in fake news detection.

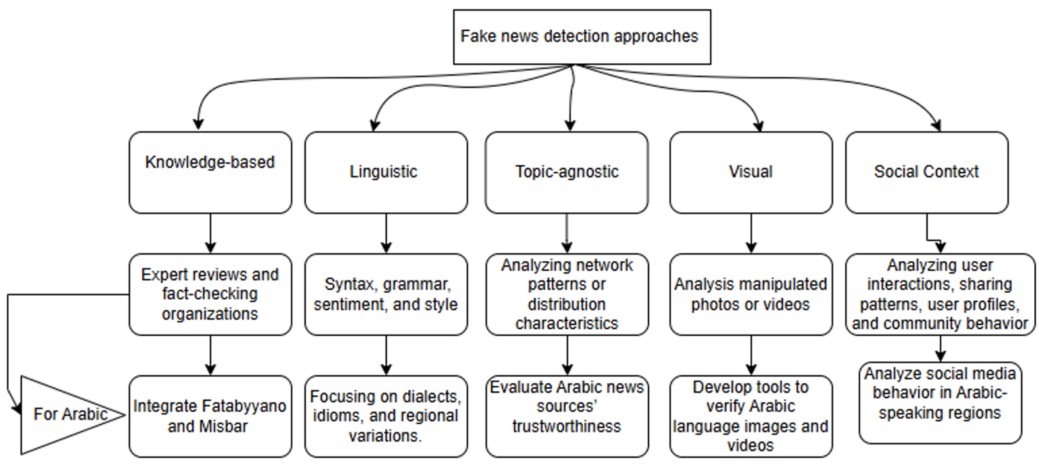

**Figure 5 Fake news detection approaches with attention to Arabic.**

# MACHINE LEARNING TECHNIQUES TO DETECT FAKE NEWS

Various textual, contextual, topic, and many features of news articles can be flagged by ML algorithms (*Zhang & Ghorbani, 2020*) and these can detect the use of provocative and biased language, as well as identify false information, analyze the sources cited in news articles, and compare them with authentic sources, flagging any conflicts or inconsistencies that may indicate potentially fake news. Additionally, they analyze social media engagement and suspicious behaviors (*Chalehchaleh et al., 2024*). Supervised machine learning algorithms are extensively employed in the detection process. The efficacy of these models greatly relies on the quality of training samples, posing challenges such as data scarcity, lack of structure, single-domain focus, data imbalance, labeling issues, and noise in current fake news datasets. Unsupervised machine learning algorithms, on the other hand, are not reliant on labeled datasets but face limitations in achieving the same accuracy as supervised methods (*Rohera et al., 2022*). While unsupervised techniques have been applied to detect similarity among online fake reviews and identify duplicated online reviews, their use in detecting misleading information is still constrained. Features extracted through these techniques can be categorized into three main approaches: user or source, content, and propagation features. Examples of content-based features include TF-IDF and the popular Bag-of-Words (BOW). Predicted-based features include Word2Vec (*Mallik & Kumar, 2024*), ELMo (*Peters, 2018*), FastText (*Joulin et al., 2016*), BERT (*Devlin et al., 2018*), and AraVec (for Arabic) (*Soliman, Eissa & El-Beltagy, 2017*). TF-IDF measures word importance in a document relative to a collection. Word2Vec and GloVe create word embeddings capturing semantic relationships. ELMo generates contextualized embeddings, while FastText includes subword information. BERT provides deep contextual embeddings from transformers, and AraVec, based on Word2Vec, specializes in Arabic text. Table 3 provides an overview of various features extracted and utilized at different levels for fake news detection, highlighting the multi-faceted approach required

**Table 3 Extracted features for fake news detection at different levels.**

| Level | Extracted Features |
| --- | --- |
| Source/user level | Profile features (*Shahid et al., 2022*), user credibility; behavior feature (*Atodiresei, Tănăselea & Iftene, 2018*); publisher emotions and social emotions (*Luvembe et al., 2023*). |
| Content level | Style features (*Probierz, Stefański & Kozak, 2021*); the positivity, negativity, and similarity content (*Mahyoob, Al-Garaady & Alrahaili, 2020*); textual features (*Zhou et al., 2020*); visual features (*Umer et al., 2020*); deep fake video detection (*Yu et al., 2021*). |
| Propagation level | Network features (*Shu et al., 2017*); propagation tree, depth of propagation, impact, and popularity level (*Shu et al., 2020b*); posting behavior (*Xu et al., 2019*); propagation, temporal, and structural (*Meel & Vishwakarma, 2020*); dissemination across time (*Zhang & Ghorbani, 2020*). |

to tackle this problem effectively. Source/user-level features, such as profile credibility, user behavior, and publisher emotions, are crucial for assessing the reliability of the information's origin. These features provide insights into whether the source has a history of sharing credible content or engaging in suspicious activity. Content-level features, including textual elements, visual cues, and linguistic properties like positivity or negativity, help detect stylistic inconsistencies or manipulative narratives in the news itself. Features like deepfake detection further address challenges in identifying falsified multimedia content. Propagation-level features, such as network structures, dissemination speed, and the depth of propagation, are essential for understanding how fake news spreads across platforms. These features capture the dynamics of information flow and reveal patterns associated with viral misinformation. Together, these features form a comprehensive framework for enhancing fake news detection across multiple dimensions. Many studies have significantly contributed to the field of fake news detection in English, employing a range of advanced techniques and up-to-date ML algorithms. One such study extracted textual features and employed several ML algorithms, including decision trees (DT), log-likelihood ratio (LLR), gradient boost, and support vector machines (SVM). The features were extracted using the TF-IDF method, and the analysis of 10,700 posts and articles achieved an impressive F1-score of 93.32% (*Patwa et al., 2021*). News title features with FastText and deep learning models were introduced by *Taher, Moussaoui & Moussaoui (2022)* whose research reveals that recurrent neural networks (RNNs) outperform convolutional neural networks (CNNs) and long short-term memory (LSTM) networks. When FastText word embeddings were utilized, accuracy rates exceeded 98%. An advanced multimodal method that includes both text and images, where text features are extracted using BERT and visual features are obtained using the Swin transformer, followed by a robust fusion mechanism, surpassing other methods, and achieving an accuracy of 83.3% (*Jing et al., 2023*). The named entity recognition technique using TF-IDF with an SVM classifier achieved an accuracy of 96.74%, followed by the DT approach, which achieved high accuracy on the ISOT Fake News dataset, reaching 96.8% (*Garg, 2023*). Word2Vec combined with a stacked LSTM model effectively utilizes sentiment analysis features to detect fake news, with accuracy reaching 98.14% (*Mallik & Kumar, 2024*). In addition, a quantum multimodal fusion-based model (QMFND) was introduced for fake news detection, integrating image and textual features processed through a quantum convolutional neural network (QCNN), achieving high accuracies of 87.9% and

**Table 4  Summary of datasets utilized in fake news detection.**

| Dataset | Size | Label | Domain | Language | Year |
|---|---|---|---|---|---|
| FakeNewsNet (*Shu et al., 2020a*) | 23,921 | Fake, Real | Politics | English | 2020 |
| FNC-1 (*Umer et al., 2020*) | 75,385 | Agree, Disagree | Multi-domain | English | 2020 |
| AraNews (*Nagoudi et al., 2020*) | 5,187,957 | True, False | News | Arabic | 2020 |
| Satirical (*Saadany, Mohamed & Orasan, 2020*) | 3,185 fake articles, 3,710 real articles | Real, Fake | Political | Arabic | 2020 |
| COVID-19 (*Patwa et al., 2021*) | 10,700 | Real, Fake | Health | English | 2021 |
| AFND (*Khalil et al., 2021*) | 606,912 | Credible, uncredible, undecided | News | Arabic | 2021 |
| ANS (*Sorour & Abdelkader, 2022*) | 1,475 Real, 3,152 Fake | Real, Fake | News | Arabic | 2022 |
| MuMiN (*Nielsen & McConville, 2022*) | 21 million tweets | Misinformation, Factual | Social media data | English | 2022 |
| Clickbait (*Bsoul, Qusef & Abu-Soud, 2022*) | 3,000 | Clickbait, not clickbait | Multidomain | Arabic | 2022 |
| MULTI Fake Detective (*Bondielli et al., 2024*) | 920,054 tweets | Real, Fake | Politics | English | 2024 |
| ConFake (*Jain, Gopalani & Meena, 2024*) | 72,413 instances | True, False | Multi-domain | English | 2024 |
| Dataset (*Hashmi et al., 2024*) | 10,1665 instances | Real, Fake | Multi-domain | English | 2024 |

84.6% on two proposed datasets (*Qu et al., 2024*). Moreover, the effectiveness of detection models relies on the quality of the training samples, particularly their diversity and size. Many studies have significantly contributed to the field of fake news detection in English, employing a range of advanced techniques and up-to-date ML algorithms. One such study extracted textual features and employed several ML algorithms, including DT, LLR, gradient boost, and SVM. The features were extracted using the TF-IDF method, and the analysis of 10,700 posts and articles achieved an impressive F1-score of 93.32% (*Patwa et al., 2021*). However, current datasets come with a set of challenges and limitations, such as the need for more inclusive labeling, potential biases, and limited coverage of certain topics or social media platforms. Nevertheless, they pave the way for future dataset creation, including the integration of more diverse sources, the identification of emerging fake news trends, and improvements in annotation quality to enhance model accuracy and generalizability. Table 4 outlines several other datasets varying in size and domain.

Table 5 demonstrates significant variation in the performance of different methods in fake news detection, largely due to the model types, dataset sizes, and specific language characteristics addressed by each approach. Transformer-based models such as BERT, AraBERT, and CAMeLBERT tend to perform well, with accuracies often exceeding those of traditional ML methods. This is because transformers are designed to capture contextual relationships effectively, which is particularly valuable for the Arabic language, where context and morphological variations are challenging to encode. For instance, the high accuracy of the hybrid Transformer models in *Mujahid et al. (2023)* and *AlEsawi & Al-Tai (2024)* (96% and 90–96%) can be attributed to their ability to leverage context more deeply than traditional methods. In comparison, models like MADAMIRA in *Nagoudi et al. (2020)* achieve a 70.06 F1 score, likely due to the simpler morphological

**Table 5 Literature review of methods used for fake news detection.**

| Works | Features | Methods | Dataset | Accuracy | Language | Year |
|---|---|---|---|---|---|---|
| *Nagoudi et al. (2020)* | Morphological analysis | MADAMIRA (https://aclanthology.org/L14-1479/) | 122K of AraNEWS and ATB (https://catalog.ldc.upenn.edu/LDC2010T08) | 70.06 F1 score | Arabic | 2020 |
| *Verma et al. (2021)* | Linguistic | SVM, BERT, CNN with TF-IDF | 72,000 articles | 96.73% | English | 2021 |
| *Najadat, Tawalbeh & Awawdeh (2022)* | Textual | LSTM, hybrid CNN-LSTM | 422 Claims and 3,042 articles | 68.2%, 70% | Arabic | 2022 |
| *Shishah (2022)* | Linguistic | jointBERT, Qarib, AraGPT2, and AraBERT | Covid19Fakes, AraNews, Satirical News, ANS Datasets | AraGPT2 achieved 88% accuracy | Arabic | 2022 |
| *Elaziz et al. (2023)* | Contextual | AraBERT with Fire Hawk Optimizer FHO (https://link.springer.com/article/10.1007/s10462-022-10173-w) | Three datasets related to COVID-19 | 72% | Arabic | 2023 |
| *Mujahid et al. (2023)* | Contextual | Hybrid model Transformer-based RoBERTA, BERT | 27,780 unstructured tweets | 96.02% | Arabic | 2023 |
| *Wotaifi & Dhannoon (2023)* | Textual | Hybrid model Text-CNN and LSTM | (AraNews dataset, 6,796 real and 6,654 fake) | 0.914% | Arabic | 2023 |
| *Himdi & Assiri (2023)* | Linguistic | RF, LR | 544 real and fake articles | 77.2%, 69.9% | Arabic | 2023 |
| *Truică, Apostol & Karras (2024)* | Social and Textual Context | RNN, CNN | BuzzFace, Twitter15, and Twitter16 | 79.62% | English | 2024 |
| *Shaker & Alqudsi (2024)* | Textual | Text-CNN | English translated dataset (5,000) and Arabic news dataset (1,000) instance | 86.2%, 99.67% | Arabic | 2024 |
| *AlEsawi & Al-Tai (2024)* | Contextual | BiLSTM, LSTM, BERT, AraBert | AraNews and ArCovid19 Rumors datasets | 96% on ArCovid19 Rumors dataset, 90% on AraNews | Arabic | 2024 |
| *Mohamed et al. (2024)* | Contextual | AraBERT, SVM, NB, LR, and RF | COVID-19 Datasets and 1,862 tweets related to Syria war | Best accuracy LR with 86.3% | Arabic | 2024 |
| *Azzeh, Qusef & Alabboushi (2024)* | Contextual | AraBERT, CAMeLBERT, ARBERT, AraVec | news websites wiht 3,460 tweets (2,121 fake, 1,339 real) | Best accuracy with CAMeLBERT 71.3% | Arabic | 2024 |
| *Othman, Elzanfaly & Elhawary (2024)* | Contextual | AraBERT, GigaBERT, MARBERT with 2D-CNN | ANS, AraNews, Covid19Fakes | Accuracy 71.42% | Arabic | 2024 |

approach and reliance on linguistic features that may not capture nuanced context as effectively.

Dataset size is also crucial: larger datasets contribute to more generalized models and better accuracy. Smaller datasets, like the 544 samples in *Himdi & Assiri (2023)*, yield lower accuracies (69.9–77.2%) as they provide less information for training robust models. The

Fire Hawk Optimizer in *Elaziz et al. (2023)* is another factor that enhances performance, improving the performance of AraBERT on COVID-19 datasets with 72% accuracy, highlighting the impact of optimization techniques in fine-tuning. Hybrid and deep learning architectures, like CNN-LSTM, used in *Najadat, Tawalbeh & Awawdeh (2022)*, also show promising results by combining strengths from multiple architectures, though their effectiveness is still influenced by the quality and size of the data. Techniques such as CAMeLBERT and AraBERT, tailored to Arabic contexts, demonstrate that Arabic-specific embeddings often yield better performance by addressing the language's unique linguistic features. This underscores the importance of selecting models and embeddings that are well-suited for the task and language at hand to achieve higher accuracy in fake news detection.

## Arabic fake news detection

The Arabic language presents challenges for natural language processing due to its features, including intricate grammar, dialectal differences, limited data availability, resource constraints, and complex morphology. For instance, the same morphological Arabic words can convey different meanings depending on their position and the diacritic marks used in the sentence (*Nassif et al., 2022*). Arabic is written from right to left and is considered a semantic language, rich in morphological structures. It consists of 28 letters, of which three are considered vowels: (أ, و, ي). Diacritical marks are placed on words to indicate pronunciation, providing a specific meaning to each word. For example, the word (عُقد) with a ḍammah on the first letter means "necklace," while (عَقد) with a fatḥah on the same letter means "contract." Despite having the same letters, the meanings differ due to the type of diacritical mark used. Thus, Arabic often presents variations in meaning depending on the diacritical marks. The complexity of Arabic grammar and its rich semantics adds further complications, and with approximately 10,000 independent roots (*Elkateb et al., 2006*), semantic analysis becomes significantly harder. Arabic sentences can be nominal (subject–verb), like الولد أكل التفاحة (Al-walad akala at-tuffaha/"The boy ate the apple"), or verbal (verb–subject), like أكل الولد التفاحة (Akala al-walad at-tuffaha/"Ate the boy the apple") with a free word order. This variability contrasts with the fixed subject–verb order in English, increasing the load on NLP techniques and ML models for better classification of fake news (*Shaalan et al., 2019*). Most words in the Arabic language are derived from what is called a "root"; some words cannot be derived and remain as they are, such as question words and pronouns. For example, words like (ملعب)/stadium, (لاعب)/ player, and (يلعب)/plays share the root (لعب), meaning "play." Other words, such as (تكنولوجيا)/technology, are borrowed from English and lack a root. Arabic also allows the addition of suffixes, prefixes, and infixes to words, often creating new meanings for the same word. For instance, the word (سعيدة), meaning "happy," contains a suffix (the last letter). By removing it, the word changes from an adjective to a noun (*Farghaly & Shaalan, 2009*). In addition, the Arabic language still presents unique challenges, such as the use of dialectal Arabic (DA) by social media users instead of formal Arabic or Modern Standard Arabic (MSA). This usage, especially on social media, does not follow specific Arabic language rules and is instead used in a disorganized manner, often containing spelling

errors and high noise. Additionally, it is highly diverse, with dialects varying widely between Arab countries, and even within the same country, there are multiple dialects between the north, center, and south. This results in a wide variety of potential words used in spreading fake content. Dialectal variation further complicates semantic analysis, introducing differences between MSA and regional dialects (*Azzeh, Qusef & Alabboushi, 2024*). For instance, "car" is سيارة, مركبة (Markaba or Syarah) in MSA, and عربية (Arabeya) in Egyptian dialect or كرهبه (Karhba) in Tunisian. This causes semantic ambiguities, which require further investigation into various NLP solutions such as context-aware embedding models (*Habib et al., 2021*), dialect-specific lexicons (*Bouamor et al., 2018*), and advanced disambiguation techniques to accurately interpret and process the meaning in different contexts such as MADAMIRA, a morphological analysis and disambiguation tool (*Nagoudi et al., 2020*).

### Prepossessing

Pre-processing Arabic text involves converting raw data into a suitable format for advanced classification. The quality of pre-processing is often an indicator of excellent results and performance for classification algorithms (*El Kah & Zeroual, 2021*). The process begins with an initial data review, followed by data cleaning, which includes removing missing values, punctuation, duplicate letters, numbers, spaces, non-Arabic digits, symbols, and non-Arabic words. Additionally, common Arabic stop words (https://github.com/mohataher/arabic-stop-words) are removed, along with diacritics (https://en.wikipedia.org/wiki/Arabic_diacritics). Diacritics include: (Tanween Fatha) ( ً ), (Tanween Damma) ( ٌ ), (Tanween Kasra) ( ٍ ), (Fatha) ( َ ), (Damma) ( ُ ), (Kasra) ( ِ ), (Sukun) ( ْ ) and (Shadda) ( ّ ). These marks are placed above or below letters to indicate short vowels, pronunciation, or emphasis. Repeated letters such as (ااااا) are reduced to a single occurrence. Normalizing Arabic letters such as (أ), (إ), and (آ) are changed to (ا). The letter (ة) is changed to (ه), and the letter (ى) is replaced with (ي) (*Soliman, Eissa & El-Beltagy, 2017*), as it reduces orthographic variations and ensures consistent text representation. This normalization minimizes noise and ambiguity, thereby enhancing model effectiveness in tasks like classification and search by treating similar forms uniformly. Tokenization is then performed on the data by splitting text into tokens. For example, the Arabic sentence فاز المنتخب الأقوى في المباراة would be tokenized into separate words: "فاز", "المنتخب", "الأقوى", "في", "المباراة". Some pre-processing operations involve stemming, which returns words to their original root using various techniques, such as Tashaphyne (https://pypi.org/project/Tashaphyne/): Arabic Light Stemmer (*Matti & Yousif, 2023*). Lemmatization refers to returning words to their root forms when they are identical both morphologically and semantically, unlike stemming, which reduces words to their root independently and without considering the meaning. Techniques like stemming, lemmatization, and stop-word removal yield better results in Arabic text classification, especially when used together (*El Kah & Zeroual, 2021*). Figure 6 illustrates the general steps for preprocessing Arabic text, which are crucial for effective fake news detection. The process begins with data cleaning through an initial data review, followed by the removal of missing values, punctuation, numbers, non-Arabic characters, diacritic marks,

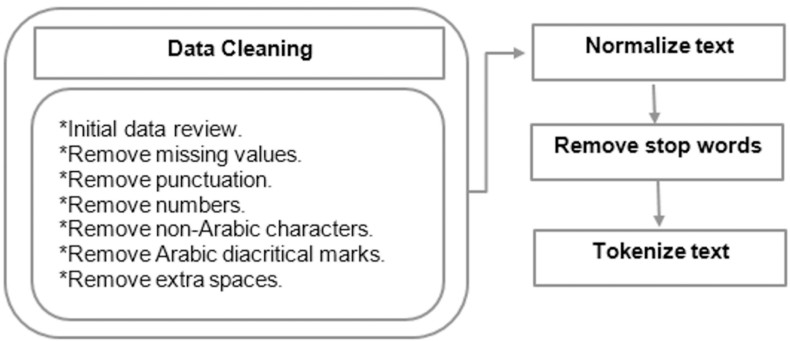

**Figure 6** Steps for Pre-processing Arabic text.               

and extra spaces to ensure consistency and reduce noise. Normalization addresses variations in writing styles, while stop word removal eliminates non-informative words. The final step, tokenization, segments the text into smaller units for analysis. These steps are particularly important for Arabic compared to English, as its complex morphology, extensive use of diacritics, and dialectal variations can obscure linguistic patterns critical for classification models. Effective preprocessing ensures the input data is clean, uniform, and representative of Arabic's unique linguistic features, enhancing the model's capacity to accurately detect fake news.

## Overview of classification models

Detecting fake news in Arabic heavily relies on manual human detection through fact-checking organizations. Facebook, for example, collaborates more with non-Arabic fact-checking organizations than with Arabic ones to control and combat fake news. However, Facebook's content management, powered by artificial intelligence algorithms, may not accurately interpret Arabic posts, leading to an increased spread of misleading content in the Arab world (*Fakida, 2021*). Classical machine learning has been widely utilized in identifying fake news, as it is used to classify textual data based on extracted features such as linguistic, stylistic, and contextual features. By pre-processing the text and training models on labeled datasets, these methods can effectively predict the credibility of new news articles. Three ML classifiers were utilized to analyze rumors (4,079 instances) surrounding the deaths of three Arab celebrities. The SVM algorithm demonstrated high accuracy, with 95.35% (*Alkhair et al., 2019*). Later updates to the dataset incorporated deep learning techniques, revealing that SVM still performed well, achieving an accuracy of 92.09%, while BiLSTM surpassed other deep learning methods in the study. It was observed that deep learning algorithms might encounter challenges in accurately classifying smaller datasets (*Alkhair, Hocini & Smaili, 2023*). Amidst the COVID-19 pandemic, a study utilizing the X streaming API with one million Arabic tweets (2,000 tweets) classified into real and fake information based on three ML classifiers (LR, SVC, and NB), achieving an accuracy of 84% (*Alsudias & Rayson, 2020*). NLP techniques and ML models, with Harris Hawks Optimizer (HHO), were utilized in combination with user, content, and linguistic features, with the best result scoring 82% using LR with TF-IDF (*Thaher et al., 2021*). RF outperformed other ML models in detecting misinformation

related to cancer treatment based on TF-IDF features and six ML models. Deep learning models, on the other hand, utilize neural network architectures like RNNs, CNNs, and Transformer-based models (*e.g.*, BERT, GPT) to extract textual features. Such models capture complex semantic patterns and relationships. The work by *Najadat, Tawalbeh & Awawdeh (2022)* uses LSTM-CNN as a classification model for classifying article headlines as fake or real. A hybrid model combining Text-CNN and LSTM was used to train on 13,450 instances (6,796 real and 6,654 fake), achieving an improved accuracy of 0.914 (*Wotaifi & Dhannoon, 2023*). Transformers approaches are particularly well-suited for Arabic fake news detection, given the large and diverse nature of the dataset. The work by *Nagoudi et al. (2020)* detected fake news using AraBERT and mBERT, where 10,000 news articles were utilized to extract POS features, achieving a 0.70 F1-score. Compared to traditional machine learning approaches like decision trees (DT), random forest (RF), linear support vector (LSV), and Naive Bayes (NB), which rely heavily on manually crafted features, transformer-based models demonstrate superior ability to capture semantic and contextual nuances in Arabic text. For instance, Mini-BERT outperformed DT, RF, LSV, and NB with 98.4% accuracy, showcasing the effectiveness of transformers in leveraging pre-trained embeddings and contextual understanding for enhanced detection. Optimization techniques further amplify the benefits of transformer-based models. The Fire Hawk Optimizer in *Elaziz et al. (2023)* significantly improved the performance of AraBERT on COVID-19 datasets, achieving 72% accuracy, highlighting the role of fine-tuning in adapting models to domain-specific datasets. Hybrid and deep learning architectures also provide a compelling comparison. For example, CNN-LSTM models used in *Najadat, Tawalbeh & Awawdeh (2022)* and *AlEsawi & Al-Tai (2024)* combined with contextual embeddings achieved high accuracy rates, such as 96% on the ArCovid19 Rumors dataset and 90% on AraNews. These hybrid models integrate sequential and spatial pattern recognition capabilities, offering a more nuanced understanding compared to traditional models. Furthermore, the use of advanced transformers like jointBERT, Qarib, AraGPT2, and AraBERT in *Shishah (2022)* further underscores the advantages of contextualized embeddings. AraGPT2 achieved 88% accuracy on the Covid19Fakes dataset, demonstrating the model's capability to generate and understand nuanced text. Additionally, *Alawadh et al. (2023)* created a large, annotated Arabic fake news *corpus* that captures various dialects and cultural contexts, employing machine learning models on fine-tuned language models such as AraBERT, CAMeLBERT, and AraVec. Their findings, supported by *Azzeh, Qusef & Alabboushi (2024)*, highlight CAMeLBERT as the most effective in generating accurate text representations.

## FAKE NEWS DETECTION CHALLENGES
### Datasets
The availability of optimal datasets in the Arabic language remains limited, and the existing ones are not as accessible to researchers compared to non-Arabic datasets. Datasets are often limited to short tweets or a single domain, such as politics or health. The annotation process is sometimes unclear, with unbalanced classes, and the cleaning and pre-processing phase is not always sufficient. Moreover, there is a need for datasets that

leverage Modern Standard Arabic (MSA) alongside Arabic dialects, with cross-dataset analysis, to enhance model accuracy and achieve generalization (*Tommasi & Tuytelaars, 2015*). To address this concern, researchers have developed Arabic datasets (*Nagoudi et al., 2020*). Investigators such as *Righi et al. (2022)* and *Shaker & Alqudsi (2024)* have adeptly translated high-quality English datasets into Arabic, allowing access to a broader range of labeled datasets. Researchers use careful translation and adaptation techniques to preserve linguistic and contextual accuracy, ensuring that these datasets maintain their integrity in Arabic while supporting cross-lingual analysis. Data augmentation (DA) techniques, such as synonym substitution, back-translation, paraphrasing, and style transfer, improve identification performance by exposing models to diverse language expressions. Generative adversarial networks (GANs) further enhance robustness by generating realistic fake news samples. However, challenges remain in ensuring the authenticity of generated samples, avoiding biases, and maintaining a balance between synthetic and real data to preserve dataset integrity and reliability (*Kuntur et al., 2024*). Data augmentation (DA) techniques have proven valuable in providing more generalized datasets and improving model robustness for Arabic fake news detection (*Mohamed et al., 2024*). Additionally, establishing partnerships with news and social media outlets provides access to authentic, well-balanced Arabic content for annotation and analysis (*Nagoudi et al., 2020*). Other strategies include using crowdsourcing to annotate data with diverse dialects, as presented by *Alsarsour et al. (2018)*, merging datasets across domains (*e.g.*, politics, health) to build richer, multi-domain resources that improve model generalization (*Khalil et al., 2021*). Future efforts should prioritize incorporating dialectal diversity into dataset development by including MSA alongside various regional dialects. Crowdsourcing can be employed for annotation, ensuring linguistic accuracy and cultural relevance through diverse annotator participation. Additionally, collaboration with news and social media outlets can facilitate access to authentic, well-balanced content, while leveraging advanced data augmentation techniques and cross-domain merging will further enhance dataset richness and model generalization.

## Fake news early prediction

In Arabic-speaking countries, social media behavior and platform differences significantly impact the spread of fake news, making it essential to tailor detection models to these unique dynamics. Platforms such as WhatsApp, Facebook, Twitter, and Instagram play crucial roles in content dissemination (*Agrawal & Sharma, 2021*), with WhatsApp being a primary channel for news sharing in private, group-based contexts. Users often share unverified content, influenced by personal beliefs and social networks, amplifying misinformation. The linguistic diversity of Arabic dialects and regional cultural nuances further complicate detection, as content can take on different meanings across various dialects. Fake news spreads rapidly, often with little fact-checking, making early detection crucial to prevent harm. Early signals of fake news, such as questionable sources, unprofessional headlines, or unreliable content, should be flagged early on to prevent widespread dissemination (*Cavalcante et al., 2024*). Researchers indicate that personal information, for example, is the most significant indicator for early fake news detection

(*Liu & Wu, 2020*). Given the fast-paced and heterogeneous nature of misinformation on social media, it is critical to develop automated systems capable of detecting fake news at an early stage. Analyzing the propagation of fake news over time and considering the reputation of publishers is a promising approach for early identification (*Algabri et al., 2024*). By incorporating measures like removing malicious accounts and providing fact-checking information, we can better control the spread and mitigate the impact of fake news. Thus, early prediction, propagation analysis, and half-truth detection are essential areas of research for tackling misinformation before it reaches a broader audience (*Zannettou et al., 2019*).

## Feature extraction

Dynamic content that includes text, audio, images, links, and symbols poses significant challenges for feature extraction and classification models, various feature extraction methods are employed for numerical representation, such as TF-IDF, bag of words (BOW), and word embeddings like Word2Vec. Advanced contextual embeddings from transformers like BERT further enhance model performance (*Shishah, 2022*; *Algabri et al., 2024*; *Mujahid et al., 2023*). In Arabic text processing, models like AraVec, trained on data from Twitter, the Arabic Web, and Wikipedia articles, have shown strong results. Additional Arabic-specific embeddings, such as Mazajak (*Farha & Magdy, 2019*), trained on large datasets of tweets, also perform well in representing the language. Meanwhile, deep learning methods, including CNN, RNN, LSTM, and CNN-LSTM, bypass manual feature engineering by automatically learning and extracting relevant features from raw data, achieving effective representation of Arabic text (*Mohamed et al., 2024*). Furthermore, fake news detection models typically rely on feature extraction using independent approaches, such as linguistic or contextual methods. However, combining multiple feature extraction techniques has proven to yield promising results (*Sahoo & Gupta, 2021*; *Almarashy, Feizi-Derakhshi & Salehpour, 2023*), as these approaches complement each other and enhance classification accuracy.

## FUTURE RESEARCH DIRECT

Most research on fake news detection predominantly focuses on English, leaving low-resource languages, including Arabic, significantly under explored. Addressing this gap involves several critical steps. First, developing comprehensive, well-annotated, and multi-domain datasets that encompass MSA, diverse dialects, and internal variations is essential. Updating existing datasets is equally important to align with evolving linguistic and social trends. The use of generative adversarial networks (GANs) for creating synthetic datasets is crucial for augmenting training data, particularly given the scarcity of annotated datasets. Enhancing pre-processing techniques to minimize noise and refining word embeddings, especially through transformer-based models, are fundamental to achieving higher classification accuracy. Feature engineering should leverage deep learning to extract linguistic patterns specific to Arabic news, taking into account the language's complexities. Integrating deep learning algorithms with other approaches and employing fine-tuning

techniques can further enhance detection performance. Utilizing advanced Arabic transformer models, such as AraBERTv2, MARBERT, and CAMeLBERT, is essential to harness their strengths and capabilities. Combining the power of these models with complementary techniques enables a more robust and accurate detection process. Moreover, employing large models like ChatGPT for detecting fake news at the linguistic level and identifying deepfakes involving images and videos is increasingly important. Understanding the factors driving the rapid dissemination of fake news on social media and examining the role of fact-checking organizations in the Arab world are also vital. Future efforts should prioritize designing hybrid fact-checking systems that integrate machine learning with human expertise and developing applications across mobile, web, and gaming platforms to raise awareness and educate users on combating fake news effectively. Additionally, exploring unsupervised learning techniques, such as clustering analysis for homogeneous groups, news sources, and authors, as well as semantic similarity analysis for news published across multiple outlets, can further enhance Arabic fake news detection capabilities (*Zhang & Ghorbani, 2020*).

## CONCLUSION

In conclusion, this article has explored the multifaceted challenge of fake news detection, highlighting its profound impact on individuals, institutions, and societies. It has examined the characteristics, domains, and life cycle of fake news, along with strategies to mitigate its spread, particularly on social media platforms. The study underscores the pivotal role of machine learning, deep learning, and transformer-based techniques in combating fake news. By categorizing detection approaches and reviewing advancements in Arabic-specific methodologies, this research contributes to the enhancement of detection systems and emphasizes the importance of developing robust datasets. Address the complex challenges in Arabic fake news detection, several actionable steps can guide future research and practical implementation. Establishing comprehensive and updated datasets that reflect Modern Standard Arabic and its diverse dialects is essential. Collaborating with linguistic experts and leveraging real-time data sources will help ensure these datasets remain relevant. To overcome the scarcity of annotated data, integrating GANs for synthetic data generation offers a practical solution to augment training resources. Enhancing pre-processing techniques and adopting advanced transformer-based word embeddings like AraBERTv2 will significantly improve classification accuracy. Employing hybrid approaches that combine machine learning algorithms with human expertise can result in more reliable fact-checking systems. Additionally, developing user-friendly applications across various platforms can raise awareness and educate users on effectively identifying and combating fake news. Lastly, exploring clustering and semantic similarity analysis can reveal patterns across news sources, paving the way for more scalable and robust detection systems. Future research is essential for refining these methods and addressing the ever-evolving nature of misinformation, particularly in Arabic content.

### Funding

The authors received no funding for this work.

### Competing Interests

The authors declare that they have no competing interests.

### Author Contributions

- Eman Salamah Albtoush conceived and designed the experiments, performed the experiments, analyzed the data, performed the computation work, prepared figures and/or tables, authored or reviewed drafts of the article, and approved the final draft.
- Keng Hoon Gan analyzed the data, authored or reviewed drafts of the article, and approved the final draft.
- Saif A. Ahmad Alrababa analyzed the data, authored or reviewed drafts of the article, and approved the final draft.

### Data Availability

This is a literature review.

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
