# Peer review of "Fake news detection: state-of-the-art review and advances with attention to Arabic language aspects"

_PeerJ Computer Science, doi:10.7717/peerj-cs.2693_

## Round 0.1 · original submission · Major Revisions

Dear Authors,

Thank you for submitting your Literature Review article. Feedback from the reviewers is now available. It is not recommended that your article be published in its current format. However, we strongly recommend that you address the issues raised by the reviewers and resubmit your paper after making the necessary changes. Before submitting the paper following should also be addressed:

1. The Introduction section should contain a well-developed and supported argument that meets the goals set out. This section should adequately introduce the subject and make it clear who the audience is and what the motivation is.
2. Please provide a clearly defined research question for this literature review paper; Differences from the review articles published within this topic should also be provided.
3. Clearly reported, reproducible, and systematic methods should be provided in order to identify, select, and critically appraise relevant research.
4. The Abstract should be attractive and contain motivation.
5. How this review paper will contribute to the scientific body of knowledge should be clearly mentioned.
6. The coverage (both temporal and domain) of the literature and how the literature was distributed across time domains should be clearly provided.
7. The document requires rigorous English proofreading, as it contains numerous grammatical errors, including those in the Abstract section.
8. The Reviewer 2 has requested that you consider specific references. You are at liberty to include them should you consider them pertinent and beneficial. However, you are under no obligation to do so, and the absence of these references will not affect the decision.

Best wishes,

Reviewer 1 ·

Basic reporting

The paper discusses the importance of detecting fake news, particularly on social media platforms, due to its potential for causing global harm. It emphasizes the unique challenges of detecting fake news in the Arabic language, including the rich morphology, complex grammatical structures, diverse dialects, and lack of annotated datasets for Arabic. The paper lacks novelty i.e. the paper largely synthesizes existing research on fake news detection without providing significant new contributions. Although it aims to focus on Arabic, much of the content reiterates findings that are already well-documented, with limited innovation or fresh insights. The methodological advances specifically targeting Arabic fake news detection remain underexplored.
The abstract should be revised to provide a clearer, more succinct overview of the paper’s scope, objectives, and key findings. Currently, it feels a bit disjointed. A more structured abstract would help readers quickly understand the contributions of the paper. The introduction could benefit from more focus. While it discusses the importance of fake news detection, it can more clearly articulate the research problem and highlight what gap the paper intends to fill in the current body of research.

Experimental design

While the paper mentions the challenges of Arabic's linguistic complexity, it does not provide concrete solutions or innovative methods to address these. The research focuses heavily on the difficulty of dialects and morphology but lacks a clear framework or novel approach for handling these challenges. Existing techniques like pre-processing are mentioned, but the paper does not explore how these techniques can be optimized or extended for Arabic text.
The discussion of Arabic datasets emphasizes their scarcity but offers little in terms of overcoming this limitation. Although the authors highlight the necessity of multi-domain and diverse datasets, the paper does not present a strategy for how such datasets could be collected or generated, nor does it discuss any work done to expand the Arabic fake news dataset landscape beyond what is already known.

Validity of the findings

The examination of machine learning techniques is somewhat superficial. For instance, while various models are listed, such as SVM, BERT, and LSTM, the paper does not delve deeply into why certain models outperform others in the context of Arabic text. The explanations for why a particular approach might be more suitable for Arabic, given its linguistic features, are underdeveloped.
While the paper discusses fake news spread through social media and the importance of social context, it doesn’t analyze how specific Arabic-speaking platforms or user behavior might influence the effectiveness of fake news detection models. Social context approaches are discussed at a high level but without sufficient attention to regional platforms or unique propagation patterns in Arabic-speaking countries.
The paper lacks empirical evidence to back up many of its claims. There is no discussion of implementation details, performance metrics, or experiments that test the effectiveness of fake news detection systems specifically for Arabic content. Including performance evaluations, benchmarks, and comparisons to state-of-the-art approaches would strengthen the paper's credibility.
Although the paper acknowledges the linguistic complexities of Arabic, such as morphology and dialectal variations, it does not provide a structured or deep discussion of how these complexities can be systematically tackled. There is limited exploration of advanced NLP techniques that could handle these issues, leaving the treatment of Arabic language aspects underdeveloped.

Additional comments

The paper could benefit from better clarity and grammatical precision. There are several grammatical errors and awkward phrases throughout the text that affect readability. For example:
1- "This paper conduct the characteristic" should be "This paper conducts a review of the characteristics."
2- "The paper provide an overview" should be "The paper provides an overview."
3- Sentences like "Arabic language presents a unique challenges in fake news detection researches" should be revised to "The Arabic language presents unique challenges in fake news detection research."
4- Rewriting some of these sentences for better flow and consistency would improve the paper’s overall readability and professionalism.
5- Ensure consistency when referring to key terms and concepts. For instance, in some places, the term "fake news" is used interchangeably with "misleading content," and these terms should be clearly defined and used consistently. The paper uses different notations for the same terms (e.g., “Fake news” vs. “fake news”). Standardizing these will improve the professional quality of the paper.
6- The conclusion could be strengthened by summarizing the key contributions of the paper in a more structured manner. Currently, it feels a bit abrupt and general. A stronger conclusion would highlight the major findings and re-emphasize the importance of addressing the challenges in Arabic fake news detection.

·

Basic reporting

1. The authors need some help with their English grammar. also, the authors need to add more references to cover most of the types of fake news detection algorithms
use in the sentence
2. Some linking words, like "to" and others, are missing throughout the paper, so it would be helpful for the researcher to review the text for language accuracy.
3. there are some missing spaces between words, and some words are stuck together.
Like "citenassif2022arabic in section 4.2 and more across the article
in section 5.1, I think that " are accessible to researchers" is are not accessible to researchers
ones are accessible to researchers compared to non-Arabic datasets.

Experimental design

For sufficient detail of the method, an enrichment of the different forms of detection (e.g., knowledge-based, linguistic, social context) by more explanation and by adding some visualization of detection techniques like flowcharts or diagrams for better illustration to clarify the type of complexity of each approach and show how an approach resolves certain aspects of how to deal with fake news detection, especially for Arabic Language. The coverage of the subject needs to be included, illustrating quantitative data to emphasize the lack of quality Arabic. The authors should add a table comparing dataset sizes, annotation quality, and domains (e.g., political vs. health-related) between Arabic and non-Arabic datasets, as this can describe the gaps in the availability of Arabic datasets. Not all sources are adequately cited especially in sections 4.2.1 and 5.3 the authors should add a citation. please check section 4 since all the table numbers are wrong and their order is also wrong.

Validity of the findings

1. More Enhancing on Arabic Specificity Computation: the authors need to add a section that compares Arabic-designed solutions with other languages to show these differences and justify the tasks of fake news detection in the Arabic language context.

2. Quantify Dataset Shortcomings: Use quantitative data to emphasize the lack of quality Arabic. I suggest that the authors add a table comparing dataset sizes, annotation quality, and domains (e.g., political vs health-related) between Arabic and non-Arabic datasets, as this can describe the gaps in the availability of Arabic datasets.

3. Many future directions have already been explored, so please structure this as a guide to help new researchers focus on specific paths that can improve fake news detection results.

Additional comments

1. The authors must ensure that all model performances (SVM, BERT, CNN) are assessed using the
same metrics (accuracy or F1 -score) in Arabic and non-Arabic fake news detection studies.
2. Add more up-to-date studies, especially regarding Arabic fake news detection, so the subject can be more approached. Emphasize best-performing methods and include caution or suggest improvements for underperforming methods. I suggest adding the references below. All suggested papers are recent and in deep learning and Machine learning, which the authors should review. The suggested papers are:
- An Effective Hybrid Deep Neural Network for Arabic Fake News Detection TA Wotaifi, BN Dhannoon
Baghdad Science Journal
- AutoKeras for Fake News Identification in Arabic: Leveraging Deep Learning with an Extensive Dataset
R Matti, S Yousif Al-Nahrain Journal of Science
- Arabic Fake News Detection Using Deep Learning Nermin Abdelhakim Othman; Doaa S. Elzanfaly; Mostafa Mahmoud M. Elhawary
- Approach for Detecting Arabic Fake News Using Deep Learning Authors khalid shaker AlhityUniversity of Anbar Arwa AlqudsiCollege of Computer Sciences and Information Technology, University of Anbar, Ramadi, Iraqhttps://orcid.org/0000-0002-1487-9285
- DOI: https://doi.org/10.52866/ijcsm.2024.05.03.049
- Improving Prediction of Arabic Fake News Using Fuzzy Logic and Modified Random Forest Model
TA Wotaifi, BN Dhannoon
Karbala International Journal of Modern Science 8 (3), 477-485
- Classification of fake news using multi-layer perceptron R Jehad, SA Yousif AIP Conference Proceedings 2334
- Classification of Covid-19 fake news using machine learning algorithms
SA Yousif, R Jehad AIP Conference Proceedings 2483 (1)

---

## Round 0.2 · Minor Revisions

Dear Authors,

Thank you for submitting your revised article. Based on reviewers' comments, your article has not been recommended for publication in its current form. However, we encourage you to address the concerns and criticisms of Reviewer a by incorporating critical comparisons, actionable recommendations, and clearer methodological transparency, and to resubmit your article once you have updated it accordingly.

Warm regards,

Reviewer 1 ·

Basic reporting

1-Several sentences are awkwardly phrased, leading to reduced clarity. For instance:
Page 3: "The digital world a massive amount of unreliable data at zero cost due to its free accessibility" needs revision for fluency.
Page 12: "The letter (/) is changed to (G), and the letter (O) is replaced with (N)" is unclear without more explanation of why these steps are important.
Repetition of content occurs in multiple sections (e.g., challenges in Arabic datasets and linguistic complexities), making the narrative less concise.
2-Some sections lack transitions, making the flow between subsections abrupt.
Figures (e.g., Figure 6, preprocessing steps) are underexplained, reducing their standalone value.
Repetitive sections, such as those addressing challenges and future directions, disrupt the logical progression.
3-Some tables and figures lack sufficient detail in their captions, such as Table 3, which could better explain why certain features are critical for fake news detection. Figure quality (e.g., Figure 6) could be improved for clarity and professional presentation.

Experimental design

1-There is insufficient transparency in how the reviewed papers were selected or excluded. For instance, the criteria for prioritizing studies from 2020 onward (Section 1.1) is not elaborated beyond a vague preference for "current research."
2-There is insufficient detail on how search results were filtered, which databases were used, and how relevancy was assessed.
3-Redundant discussions (e.g., Arabic linguistic challenges repeated in multiple sections) detract from coherence.

Validity of the findings

1-While the conclusions summarize key findings, they are general and do not emphasize actionable insights or practical implications. For example, while future research is mentioned, the steps to achieve those goals are not detailed.
2-The argument lacks depth in some areas. For example, the benefits of using transformer-based models like AraBERT are mentioned but not critically compared to traditional machine learning approaches or other advanced models.
3-The discussion of unresolved questions and future directions is too general. For example, while the need for better datasets is emphasized, there is no specific discussion on how these datasets should be developed (e.g., integrating dialectal diversity, using crowdsourcing for annotation).

---

## Round 0.3 · accepted · Accept

Dear Authors,

Thank you for clearly addresing the reviewers' comments. Your paper seems sufficiently improved and ready for publication.

Best wishes,

Reviewer 1 ·

Basic reporting

Thanks for addressing my comments.

Experimental design

non

Validity of the findings

non

Additional comments

non